# USP25 regulates KEAP1-NRF2 anti-oxidation axis and its inactivation protects acetaminophen-induced liver injury in male mice

Changzhou Cai [1,6], Huailu Ma[2,6], Jin Peng[3], Xiang Shen[4], Xinghua Zhen[3], Chaohui Yu[1], Pumin Zhang [2,3,5] ✉, Feng Ji [1] ✉ & Jiewei Wang [1] ✉

Nuclear factor erythroid 2-related factor 2 (NRF2) is a transcription factor responsible for mounting an anti-oxidation gene expression program to counter oxidative stress. Under unstressed conditions, Kelch-like ECH-associated protein 1 (KEAP1), an adaptor protein for CUL3 E3 ubiquitin ligase, mediates NRF2 ubiquitination and degradation. We show here that the deubiquitinase USP25 directly binds to KEAP1 and prevents KEAP1's own ubiquitination and degradation. In the absence of *Usp25* or if the DUB is inhibited, KEAP1 is downregulated and NRF2 is stabilized, allowing the cells to respond to oxidative stress more readily. In acetaminophen (APAP) overdose-induced oxidative liver damage in male mice, the inactivation of *Usp25*, either genetically or pharmacologically, greatly attenuates liver injury and reduces the mortality rates resulted from lethal doses of APAP.

The transcription factor nuclear factor erythroid 2-related factor 2 (NRF2) is the most important regulator of cellular redox homeostasis[1]. Together with small MAF proteins (MafF, MafG, or MafK)[2,3], NRF2 binds to the antioxidant response element (ARE)-containing promoters to regulate the expression of a series of oxidation-fighting genes, such as NAD(P)H: quinone oxidoreductase 1 (*Nqo1*), glutamate-cysteine ligase catalytic subunit (*Gclc*), and glutamate-cysteine ligase modifier subunit (*Gclm*). Under normal conditions, NRF2 levels are kept low to avoid unnecessary activation of these defensive genes. This is achieved in large part through ubiquitin-mediated proteasomal degradation. The ubiquitination of NRF2 is catalyzed by Cul3[KEAP1] in which KEAP1, Kelch-like ECH-associated protein 1, is the adaptor protein for NRF2[4,5]. KEAP1 contains as many as 27 cysteine residues that are oxidized under oxidative stress. Such oxidation results in the dissociation of NRF2 and therefore stabilization of NRF2 which then enters nucleus to activate

expression of the afore-mentioned genes. Thus, KEAP1 is a sensor for oxidative stress and a gate-keeper for the expression of detoxifying genes. KEAP1 itself also undergoes CUL3-mediated[6] as well as TRIM25-mediated ubiquitination[7], and the ubiquitinated KEAP1 is thought to be degraded mainly through autophagy[8], but may also through the proteasome pathway[7]. KEAP1 degradation seems to accelerate under oxidative stress conditions to further release restrains on NRF2[9], which suggests that the KEAP1 ubiquitination process is subjected to regulation. Oxidative stress may enhance its ubiquitination, slow its deubiquitination, or both to decrease KEAP1 levels, but the factors involved in such regulation is unclear.

Drug-induced liver injury (DILI) is a common clinical problem worldwide, and the majority of acute liver failure in the United States are caused by DILI[10,11]. Acute liver injury (ALI) induced by acetaminophen (N-acetyl-4-aminophenol, APAP) overdose is the most

[1]Department of Gastroenterology, The First Affiliated Hospital of Zhejiang University School of Medicine, Hangzhou, Zhejiang 310003, China. [2]Institute of Translational Medicine, Zhejiang University School of Medicine, Hangzhou, Zhejiang 310058, China. [3]Zhejiang Provincial Key Laboratory of Pancreatic Disease, The First Affiliated Hospital of Zhejiang University School of Medicine, Hangzhou, Zhejiang 310003, China. [4]Chaser Therapeutics, Inc., Hangzhou, Zhejiang 310018, China. [5]Cancer Center, Zhejiang University, Hangzhou, Zhejiang 310058, China. [6]These authors contributed equally: Changzhou Cai, Huailu Ma. ✉e-mail: pzhangbcm@zju.edu.cn; jifeng@zju.edu.cn; jerrywang075@zju.edu.cn

common cause of DILI in the United States[12]. Many studies have shown that the APAP-induced liver toxicity is mainly caused by N-acetyl-p-benzoquinone imine (NAPQI), a reactive metabolite of APAP, which binds to liver proteins and leads to oxidative stress after rapidly depleting the cellular glutathione (GSH) pool, and eventually causes massive necrotic cell death[13,14]. The hepatocytic defense against APAP insult (and other oxidative stress) relies on NRF2. In its absence, APAP-induced damage is exacerbated extensively[15,16].

Ubiquitin-specific peptidase 25 (USP25) was initially identified to regulate anti-viral immunity through modulating TRAF proteins[17] and later found to be associated with cancer development as it helps maintain the expression levels of Tankyrase, a poly-ADP-ribosyltransferase involved in Wnt signaling[18], BCR-ABL in leukemia cells[19], and EGFR through modulating its internalization and degradation[20]. More recently, USP25 was found implicated in the inflammation and injury of intestine induced by dextran sulfate sodium salt (DSS)[21] and pancreas by cerulein[22]. Here we report that USP25 deubiquitinates KEAP1 and prevents it from excessive ubiquitination and degradation. When USP25 function is blocked through genetic knockout or pharmacologic inhibition, KEAP1 levels decrease while NRF2 levels increase, making hepatocytes better prepared to deal with oxidative insults such as APAP overdose. We further show that the USP25 inhibitor CT1113 we developed[23] could not only prevent APAP-induced liver injury, but also be used as a treatment after APAP overdose, providing a potential pharmacological means to treat liver injuries or other clinical conditions where NRF2 activation is beneficial.

## Results

### The loss of *Usp25* protects liver from APAP-induced injury

To look into the potential involvement of *Usp25* in APAP-induced liver damage, we first looked at its expression in a mouse model of acute APAP liver toxicity[24] (Fig. 1a and Supplementary Fig. 1a, b). Quantitative PCR analysis of the control and APAP-treated liver samples showed a 2-fold decrease of *Usp25* mRNA levels in the treated samples (Fig. 1b), and its protein levels decreased accordingly (Fig. 1c). Apparently, APAP treatment downregulates *Usp25* expression. This downregulation might just be a mere coincidence or it might help relieve APAP-ALI. To determine which scenario is the case, we employed *Usp25* knockout mice. We reasoned that inactivating *Usp25* preemptively (and completely) might allow the animals better prepared to fend off APAP-induced oxidative assault. The *Usp25* knockout mice along with *Usp25* wildtype littermates were subjected to the same APAP overdosing scheme (Fig. 1d), and the liver injury was assessed at 0, 2, 4, 6 and 24 h after APAP treatment. As shown in Fig. 1e, serum ALT and AST levels started to rise at 4 h and kept rising. However, they were markedly lower in *Usp25* KO mice than those in *Usp25* WT controls. Meanwhile, APAP-induced typical centrilobular necrosis in the liver as assessed by H&E staining and quantification was markedly attenuated too in the *Usp25* KO mice (Fig. 1f).

It is known that APAP induces oxidative stress[25]. We therefore used immunofluorescence to analyze and quantify the levels of reactive oxygen species (ROS) with a ROS probe dihydroethidium in the frozen liver sections from these mice. The accumulation of ROS in the *Usp25* WT mice was readily detectable after APAP treatment, but no such accumulation was observed in *Usp25*−/− mice (Fig. 1g). Moreover, the *Usp25*−/− mice were much less likely to succumb to death after a lethal dose of APAP (Fig. 1h). Collectively, these data show that APAP-induced acute liver injury was strongly attenuated in *Usp25*-deficient mice.

### USP25 regulates KEAP1-NRF2 axis in the defense against oxidative assaults

To begin exploring the mechanism(s) behind the apparent protection afforded by the absence of *Usp25*, we first examined JNK

activation which is known to be induced by APAP overdosing and its activation also provides protection[24,26]. Indeed, JNK was activated strongly at 6 h after APAP treatment in the liver. However, the degree of activation is similar between *Usp25* WT and *Usp25* KO mice (Supplementary Fig. 2a). In addition, it is known that cytochrome P4502E1 (CYP2E1) is the main enzyme for liver cells to metabolize APAP in the early stage to generate reactive intermediate NAPQI that covalently binds to proteins to cause oxidative damages[27]. Thus, it is possible that *Usp25* deficiency might reduce the expression of CYP2E1, leading to less production of the harmful NAPQI and apparent protection form APAP in the end. However, the loss of *Usp25* did not alter the protein levels of CYP2E1 nor the concentration of APAP-CYS, a marker indicating the degree of protein oxidation at 6 h after APAP administration (Supplementary Fig. 2b, c). These results indicate that the protection provided by *Usp25* deficiency is likely mediated by another mechanism(s).

We next examined the expression of NRF2, a key transcription factor in the defense against APAP overdose-induced liver injury[28]. As shown in Fig. 2a, the loss of *Usp25* increased the levels of NRF2 in the liver even before APAP treatment, and increased that further after APAP, suggesting more NRF2 activation in the absence of *Usp25*. Indeed, NRF2 target gene expression increased in the KO mice (Fig. 2b). However, this increase was not evident at time 0 (i.e., before APAP treatment), suggesting additional mechanisms are at play to keep precocious activation of NRF2 target genes when there are no real threats. Interestingly, *Keap1* mRNA levels did not show any differences between *Usp25* WT and *Usp25* KO mice, but was downregulated by APAP treatment (Fig. 2b). As expected for the increased levels of NRF2 activation, there were more anti-oxidation capacity (represented by the levels of GSH) present in the *Usp25* knockout liver at 2, 4 and 6 h after APAP treatment (Fig. 2c). Interestingly, the protein (but not mRNA) levels of KEAP1, the adaptor for NRF2 ubiquitination, decreased in the absence of *Usp25* (Fig. 2a, b). Similar decrease of KEAP1 and increase of NRF2 levels were observed in the primary hepatocytes isolated from *Usp25*+/+ and *Usp25*−/− mice treated with APAP (Fig. 2d), recapitulating what was found in vivo (Fig. 2a). In human hepatocarcinoma cell line HepG2, the depletion of *USP25* also resulted in a decrease of KEAP1 expression and an increase of NRF2 expression (Fig. 2e). We noticed that the NRF2 levels did not increase appreciably after APAP treatment in the liver (Fig. 2a), in line with other reports[29,30], but they did increase in primary mouse hepatocytes treated with APAP (Fig. 2d). The cause of the discrepancy is not clear, but it is likely a result of uneven exposure of the hepatocytes to APAP in vivo.

As an E3 ubiquitin ligase for NRF2, KEAP1 expression levels are expected to determine the ubiquitination and stability of NRF2. Indeed, we found that when *USP25* expression was depleted, NRF2 ubiquitination levels decreased significantly (Fig. 2f) and the stability of NRF2 increased (Fig. 2g). More importantly, the overexpression of *KEAP1* could overcome the effect of *USP25* depletion on NRF2 expression (Fig. 2h). Collectively, these results indicate that USP25 negatively regulates NRF2 through KEAP1.

*Usp28* is a close homologue of *Usp25*[31]. To determine if *Usp28* also plays a role in APAP-ALI, we subjected *Usp28* knockout mice generated previously in the lab[32] to APAP overdosing. Not surprisingly, *Usp28*-deficient mice behaved similarly as *Usp28* wildtype littermates. There was no alleviation of the liver injury at all (Supplementary Fig. 3a−c) and the loss of *Usp28* did not cause an upregulation of NRF2 target gene expression (Supplementary Fig. 3d). Further, the deficiency of *Usp28* did not alter the levels of KEAP1 and NRF2 in the liver (Supplementary Fig. 3e), and the depletion of *USP28* in HepG2 cells caused no effects on the expression of KEAP1 and NRF2 (Supplementary Fig. 3 f) or on the ubiquitination of KEAP1 (Supplementary Fig. 3g). Thus, *Usp28* is not involved in the response to APAP, differing from its homologue *Usp25*.

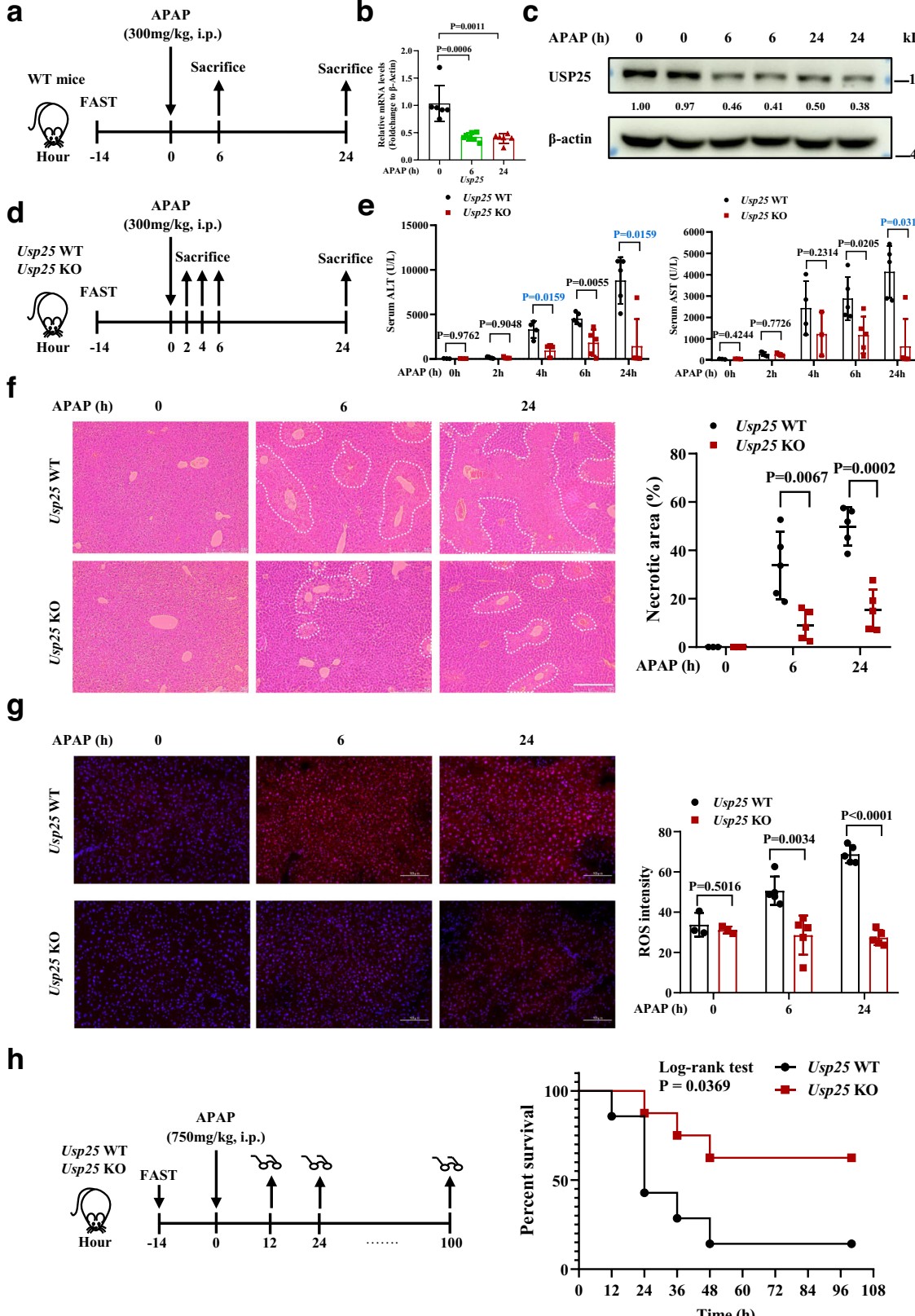

## USP25 interacts with and deubiquitinates KEAP1

In another line of inquiry into how *Usp25* regulates APAP response in the liver, we sought to identify USP25 interactors and hoped to find clues about the role the DUB plays. To that end, we overexpressed *USP25* in HEK293T cells and immunoprecipitated FLAG-tagged USP25. After extensive wash, the immunoprecipitates were separated in an SDS-PAGE gel (Supplementary Fig. 3a) and the gel was divided into several fractions for mass-spec analysis which yielded many potential USP25 interacting proteins. Much to our satisfaction, KEAP1 was found at the top of USP25 interactor list along with CUL3 (Fig. 3b). The interaction with KEAP1 was confirmed in a GST pulldown assay (Fig. 3c) and was further validated through co-immunoprecipitation

**Fig. 1 | The loss of Usp25 protects the liver from APAP-induced injury. a** The scheme of APAP treatment. Male C57BL/6 mice were fasted for 14 h, treated with APAP (300 mg/kg, i.p.), and sacrificed at 0, 6 or 24 h after APAP administration. **b** Quantification of *Usp25* expression in the liver from the animals of (**a**) via qPCR. 0 h: *n* = 6 mice, 6 h: *n* = 7 mice, 24 h: *n* = 6 mice. **c** Quantification of *Usp25* expression in the liver from the animals of (**a**) via western blotting. The numbers are normalized relative expression of USP25. **d** The scheme of APAP treatment in *Usp25*⁺/⁺ and *Usp25*⁻/⁻ male mice. The animals were fasted for 14 h, treated with APAP (300 mg/kg, i.p.), and sacrificed at 0, 2, 4, 6 or 24 h after APAP administration. **e** Serum ALT and AST levels were determined in mice from (**d**). *Usp25* WT: 0 h: *n* = 3 mice, 2 h: *n* = 5 mice, 4 h: *n* = 4 mice, 6 h: *n* = 5 mice, 24 h: *n* = 5 mice, *Usp25* KO: 0 h: *n* = 3 mice, 2 h:

*n* = 4 mice, 4 h: *n* = 3 mice, 6 h: *n* = 5 mice, 24 h: *n* = 5 mice. **f** Hematoxylin and eosin staining of the liver sections from mice in (**d**). Necrotic areas were encircled and quantified. 0 h: *n* = 3 mice, 6 h: *n* = 5 mice, 24 h: *n* = 5 mice. Scale bar, 200 µm. **g** Immunofluorescence analysis of ROS in frozen liver sections from mice in (**d**). 0 h: *n* = 3 mice, 6 h: *n* = 5 mice, 24 h: *n* = 5 mice. Scale bar, 100 µm. The fluorescent signal strength was quantified with Image J. At least 3 view fields under a 20x objective were quantified per section. **h** The treatment scheme and survival of *Usp25*⁺/⁺ and *Usp25*⁻/⁻ mice treated with APAP (750 mg/kg, i.p.). *Usp25* WT: *n* = 7 mice, *Usp25* KO: *n* = 8 mice. Error bars denote SEM. Two-tailed student's *t* tests analysis (b, e, f and g), non-parametric tests with blue colored (**e**). Source data are provided as a Source Data file.

experiments in HEK293T cells (Fig. 3d). To demonstrate that the interaction could occur in liver, we immunoprecipitated endogenous USP25 or KEAP1 from mouse liver homogenates and examined the presence of another in the precipitates. As shown in Fig. 3e, USP25 could bring down KEAP1, and vice versa. The interaction strength seemed to decrease after APAP treatment, but that could just be a reflection of the now decreased expression levels of both KEAP1 and USP25.

To further characterize the interaction between USP25 and KEAP1, we constructed a series of truncations in both proteins and expressed these fragments to determine the functional domains or regions responsible for the interaction (Fig. 3f, g). In KEAP1, we found that it was the double glycine repeat (DGR) domain, not the BTB (broad-complex, tramtrack, bric a brac) domain, nor the intervening region (IVR), that interacted with USP25 (Fig. 3f). In USP25, it was mainly the CTD (C-terminal domain) that interacted with KEAP1 (Fig. 3g).

## USP25 protects KEAP1 from ubiquitination and degradation

Given that USP25 is a deubiquitinase, we hypothesized that it protects KEAP1 from ubiquitination and degradation. In other words, USP25 functions to stabilize KEAP1. To determine if that is the case, we depleted or overexpressed *USP25* in HepG2 cells as well as in AML12 cells. The knockdown of *USP25* reduced the expression levels of KEAP1 (Fig. 4a and Supplementary Fig. 4a, b). Such a reduction, however, could be restored by a 10 µM proteasome inhibitor Z-Leu-Leu-Leu-al (MG132) treatment, suggesting KEAP1 is degraded via the proteasome (Fig. 4a). On the other hand, the overexpression of *USP25*, but not the deubiquitinase-dead *USP25-C178S*, increased KEAP1 levels (Fig. 4b). Further, the depletion of *USP25* also shortened the half-life of KEAP1 (Fig. 4c), whereas overexpression of the deubiquitinase (but not the deubiquitinase-dead mutant) extended the half-life (Fig. 4d). In line with this, KEAP1 was found more ubiquitinated in HepG2 cells depleted of *USP25* than in the control cells (Fig. 4e). In addition, there was much less ubiquitination on KEAP1 when *USP25* but not *USP25-C178S* was overexpressed (Fig. 4f). Similar results were obtained with HEK293T cells (Supplementary Fig. 4c, d). Taken together, these data demonstrate that USP25 protects KEAP1 from ubiquitination and subsequent degradation.

## The protection by the loss of Usp25 against APAP-induced liver injury is mediated through NRF2

The demonstration that USP25 positively regulates KEAP1 and therefore negatively regulates NRF2 provides an explanation for the apparent protection against APAP-induced liver injury by the absence of *Usp25*. To show that is really the case, we depleted *Nrf2* expression in the liver and looked at the impact on the protection. We generated high titer adenoviruses carrying an expressing cassette for shRNA against *NRF2* (AdshNRF2) or for scrambled control shRNA (AdshNC), and injected the viruses into wildtype or *Usp25* knockout mice via tail vein (Fig. 5a). As expected, AdshNRF2 caused NRF2 expression levels to decrease in both *Usp25*⁺/⁺ and *Usp25*⁻/⁻ mice but had no effect on KEAP1 expression (Fig. 5b). However, the NRF2

expression levels in *Usp25*⁻/⁻ mice were still much higher than that in *Usp25*⁺/⁺ controls. In fact, they reached similar levels in *Usp25*⁺/⁺ animals without *Nrf2* depletion (Fig. 5b). In line with the NRF2 expression levels, APAP-induced liver damage as indicated by the serum ALT and AST levels, was much worse in *Nrf2*-depleted wildtype animals than those in the *Usp25* knockout controls (Fig. 5c). The liver damage was also worse in *Usp25*⁻/⁻ animals injected with AdshNRF2 than that in the *Usp25*⁻/⁻ animals injected with AdshNC (Fig. 5c). The GSH levels (Fig. 5d), as well as the centrilobular necrotic area seen on histological sections of the liver tissues (Fig. 5e), followed the same trend. However, the damage levels in *Usp25* KO animals with *Nrf2* depleted only reached that of *Usp25*⁺/⁺ mice without *Nrf2* depletion (Fig. 5c–e). The depletion of *Nrf2* in *Usp25*⁻/⁻ mice made these animals look apparently like the *Usp25* WT mice in response to APAP overdosing. These results strongly suggest that the protective effects afforded by *Usp25*-deficiency is mediated through *Nrf2*.

## Pharmacological inhibition of USP25 can prevent APAP-ALI

Having demonstrated that the loss of *Usp25* could protect APAP overdosing, we next wanted to determine if pharmacologic inhibition could do the same so that a USP25 inhibitor-based treatment for APAP overdosing could be developed. CT1113, a potent USP25 inhibitor CT1113, was recently reported by our group[23]. In both HepG2 cells or AML12 cells, a 48-h treatment with CT1113 downregulated the protein levels of both USP25 and KEAP1 in a dose-dependent manner (Supplementary Fig. 5a, b), demonstrating the effectiveness of the inhibitor. To look into its effectiveness in vivo, we treated mice with CT1113 (20 mg/kg, bid) or the vehicle control through an oral gavage 48 h before the APAP administration, and the animals were sacrificed at 0, 2, 4, 6, and 24 h after APAP injection for analyses (Fig. 6a). As shown in Fig. 6b, the serum levels of ALT and AST were significantly lower in CT1113-treat mice than that in the control animals. The degree of central necrosis in the liver lobules of CT1113-treated mice was also much less severe than that in the controls (Fig. 6c). Accordingly, the levels of NRF2 target genes *Nqo1*, *Gclc*, and *Gclm* increased in the liver tissues of the mice treated with CT1113 (Fig. 6f), and the levels of GSH were also tuned up by CT1113 treatment (Fig. 6g). These results indicate that USP25 inhibitor can indeed protect APAP overdosing. However, CT1113 could not bring additional protection in *Usp25*-deficient mice (Supplementary Fig. 5c), but still could protect *Usp28*⁻/⁻ mice (Supplementary Fig. 5d). Further, CT1113 treatment did not result in any changes in the expression of CYP2E1 (Supplementary Fig. 5e). Taken together, these data strongly suggest that the small molecule works through USP25.

To determine the therapeutic potential of CT1113 on APAP-ALI, we subjected mice to APAP overdosing first and then gave them CT1113 6 h later, for a total of 3 times (every 8 h) and the mice were sacrificed at 24 h (Fig. 7a). Compared with the vehicle controls, serum ALT and AST levels in the CT1113-treated group were significantly lower (Fig. 7b). In line with that, the histology of the liver samples showed that CT1113 significantly reduced APAP-induced liver necrosis (Fig. 7c). Moreover, we wanted to determine whether CT1113 could reduce the mortality

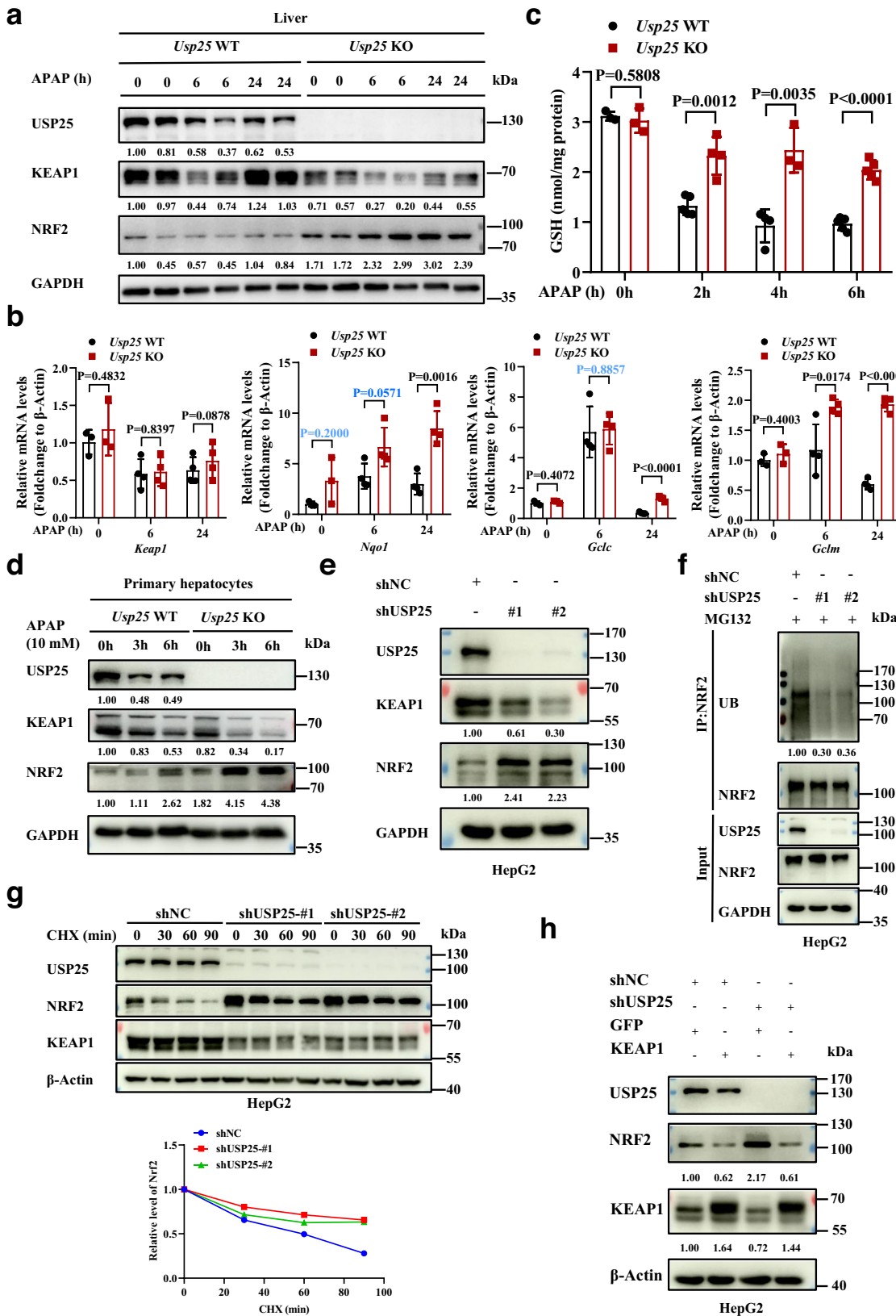

rate caused by a lethal dose of APAP (750 mg/kg) administration (Fig. 7d). As shown in Fig. 7e, the mice pre-treated with CT1113 had a much higher survival rate than the vehicle-treated animals. These data demonstrate that pharmacologic inhibition of USP25 can effectively attenuate APAP-induced liver injury and reduce mortality from lethal dose of APAP.

Next, we compared CT1113 with an electrophilic NRF2 inducer (NRF2 activator), bardoxolone methyl (RTA402)[33]. We first induced APAP overdose in mice, and then administered CT1113 or RTA402 one hour later. The mice were sacrificed at 6 h after APAP injection to assess the treatment outcomes (Supplementary Fig. 6a). Compared to the vehicle control, both CT1113 and RTA402 significantly reduced

**Fig. 2 | The activation of NRF2 pathway by USP25 deficiency. a** Western blotting analysis of liver proteins from *Usp25*[+/+] and *Usp25*[−/−] mice in Fig. 1d. **b** Quantification of NRF2 target gene expression via qPCR in the liver from the mice (*n* = 4 per group). **c** The levels of GSH in the liver from the mice in Fig. 1d. *Usp25* WT: 0 h: *n* = 3 mice, 2 h: *n* = 5 mice, 4 h: *n* = 4 mice, 6 h: *n* = 5 mice, *Usp25* KO: 0 h: *n* = 3 mice, 2 h: *n* = 4 mice, 4 h: *n* = 3 mice, 6 h: *n* = 5 mice. **d** Western blotting analysis of the indicated proteins in the primary hepatocytes isolated from *Usp25*[+/+] and *Usp25*[−/−] mice (*n* = 2 biologically independent experiments). **e** Western blotting analysis of the indicated proteins in HepG2 cells (*n* = 3 biologically independent experiments). **f** USP25 promotes ubiquitination of NRF2. *USP25* expression in HepG2 cells was knocked down with two independent shRNAs and NRF2 was immunoprecipitated and blotted for the presence of ubiquitin (*n* = 3 biologically independent

experiments). **g** Stabilization of NRF2 by *USP25* depletion. *USP25* expression in HepG2 cells was knocked down with two independent shRNAs. The cells were then treated with cycloheximide for various of time before harvested for analyses of the indicated proteins via western blotting (*n* = 2 biologically independent experiments). The relative expression of NRF2 was plotted. **h** Overexpression of *KEAP1* diminished the effects of *USP25* depletion on NRF2. *USP25* expression in 293 T cells was knocked down first and the cells were then transiently transfected with *KEAP1* or GFP expression plasmids (*n* = 2 biologically independent experiments). The indicated proteins in these cells were analyzed with western blotting. The numbers under western blotting bands represent normalized relative expression. Error bars denote SEM. Two-tailed student's *t* tests analysis (**b** and **c**), non-parametric tests with blue colored (**b**). Source data are provided as a Source Data file.

APAP-ALI, and the two agents displayed similar efficacies (Supplementary Fig. 6b, c).

## Discussion

The KEAP1-NRF2 axis is critically involved in cellular defense against oxidative stress. In this binary regulatory system, KEAP1 acts as a sensor of cellular oxidants through its many cystine residues and as an E3 for NRF2[4]. We show here that USP25 is a deubiquitinase for KEAP1, adding an additional layer of regulation to the KEAP1-NRF2 axis. Under unstressed conditions, KEAP1 brings NRF2 to CUL3 ubiquitin ligase for ubiquitination, leaving NRF2-mediated transcriptional program running at minimum. While bringing NRF2 to CUL3 complex for ubiquitination, KEAP1 might get itself ubiquitinated, making USP25 a necessary DUB to prevent KEAP1 levels from going down and that of NRF2 from going up (Fig. 7f). Interestingly, *Usp25* and *Keap1*, acting as suppressors of NRF2, are downregulated transcriptionally in response to APAP overdosing (Figs. 1b, 2b), suggesting that both of them are part of the anti-oxidation defense program in the liver. It is unclear at the moment which transcription factor is responsible for the expression of *Usp25* and *Keap1*, and how their transcription is suppressed in response to APAP treatment, but we suspect that there is a general transcription repression mechanism which is activated in response to APAP- and perhaps other agents-induced oxidative stress as well. In addition, we also don't know if the downregulation of *Usp25* is responsible for the observed decrease of KEAP1 levels (Fig. 2a), but it certainly could help in that regard.

*Usp25*-deficient mice are remarkably resistant to APAP-induced liver injury. Compromising *Nrf2* in these mice via shRNA-mediated knockdown could make them susceptible to APAP-ALI again, indicating that the effect from the absence of *Usp25* is largely mediated through NRF2. Consistent with that, NRF2 target gene expression is elevated in the absence of *Usp25* (Fig. 2b). However, the elevation of NRF2 target gene expression is not observable before APAP treatment, indicating that merely increasing the levels of NRF2 is not sufficient to activate its target gene expression. Something else must happen to fully release NRF2's capacity to activate transcription. Apparently, there are mechanisms in place to prevent precocious activation of NRF2-mediated transcription program which could be harmful if activated inappropriately. It is known that NRF2 requires small MAF proteins as a partner to activate transcription[2]. Thus, one potential mechanism could be that APAP-induced oxidative stress is a necessary signal for NRF2 to partnering up with MAF proteins and to activate transcription. *Usp25* deficiency just primes the hepatocytes for NRF2 activation.

Priming hepatocytes for NRF2 activation could also be brought about by pharmacologic inhibition of USP25. Using an USP25 inhibitor CT1113 generated by our group[23], we showed that inhibiting USP25 could produce protective effects against APAP which are near identical to that derived from genetically inactivating the DUB (Fig. 6). The protection against APAP by CT1113 is even more evident in that the compound could make mice resistant to a lethal dose of

APAP (Fig. 7d, e). Furthermore, not only could it provide protection, CT1113 could also be used afterwards as a treatment for APAP-ALI (Fig. 7a–c).

CT1113 also inhibits USP25's close homologue USP28 with a similar if not better efficacy[23]. Since CT1113 treatment provided very similar protection against APAP as that of *Usp25* knockout mice, we believe there is little contribution from inhibiting USP28 in CT1113's protective effect against APAP-ALI. Indeed, *Usp28*-deficient mice were as susceptible as their controls to APAP overdosing (Supplementary Fig. 3), and CT1113 still could reduce the liver damage in *Usp28*-deficient mice (Supplementary Fig. 5d). Thus, it is highly likely that CT1113's effect comes from inhibiting USP25 alone.

In summary, we have uncovered the function of the deubiquitinase USP25 in regulating hepatocytic oxidative defense through maintaining KEAP1 levels (Fig. 7f). The demonstration that the USP25 inhibitor CT1113 can protect as well as treat APAP-ALI in mice suggests that CT1113 could be developed as a therapeutic agent for APAP-ALI. Since the anti-oxidation function of NRF2 is beneficial in a large number of disease settings including neurological and liver diseases, our finding here opens the door for future exploration and application of USP25 inhibitors for various disease conditions.

## Methods
### Animal experiments
All animal experiments were approved by the Animal Care and Use Committee of the First Affiliated Hospital of Zhejiang University. *Usp25* knockout mice were obtained from Dr. Chen Dong at Tsinghua University, and *Usp28* knockout were generated in the lab. *Usp25*[−/−] or *Usp28*[−/−] were generated from intercrossing of corresponding heterozygous mice and the *Usp25*[+/+] or *Usp28*[+/+] mice were used as controls in the experiments. C57BL/6 J mice (6- to 8-week-old) were purchased from Hangzhou Ziyuan Laboratory Animal Technology Co., Ltd. (Zhejiang, China). All animals were housed in a SPF facility in the First Affiliated Hospital of Zhejiang University under temperature (22 ± 2°C) and humidity-controlled (55 ± 5%) conditions with 12-hour light/12-hour dark circadian cycle. They were fed with a standard laboratory diet (Research Diets, D12450K, USA) and had free access to water. To induce liver injury with APAP, male mice were used. They were fasted for 14 h and injected intraperitoneally (i.p.) with either saline or APAP (Sigma-Aldrich, Shanghai, China) at 250, 300, or 750 mg/kg body weight. The animals were anesthetized via isoflurane induction at various timepoints after APAP dosing for analyses. To treat the animals with USP25/28 inhibitor CT1113, the compound was dissolved in 40% (2-hydroxypropyl)-β-cyclodextrin (Sigma-Aldrich) at a concentration of 50 mg/ml as a stock solution and diluted with water to 2 mg/ml before use. CT1113 was administered with oral gavage at 20 mg/kg body weight or injected intraperitoneally at 10 mg/kg. To test the effect of the NRF2 inducer RTA402 (Selleck, Texas, USA) on APAP-ALI, we dissolved the chemical sequentially in DMSO, PEG300, and Tween80, and finally diluted it with ddH2O to a concentration of 2 mg/ml, with a final gavage dose of 20 mg/kg.

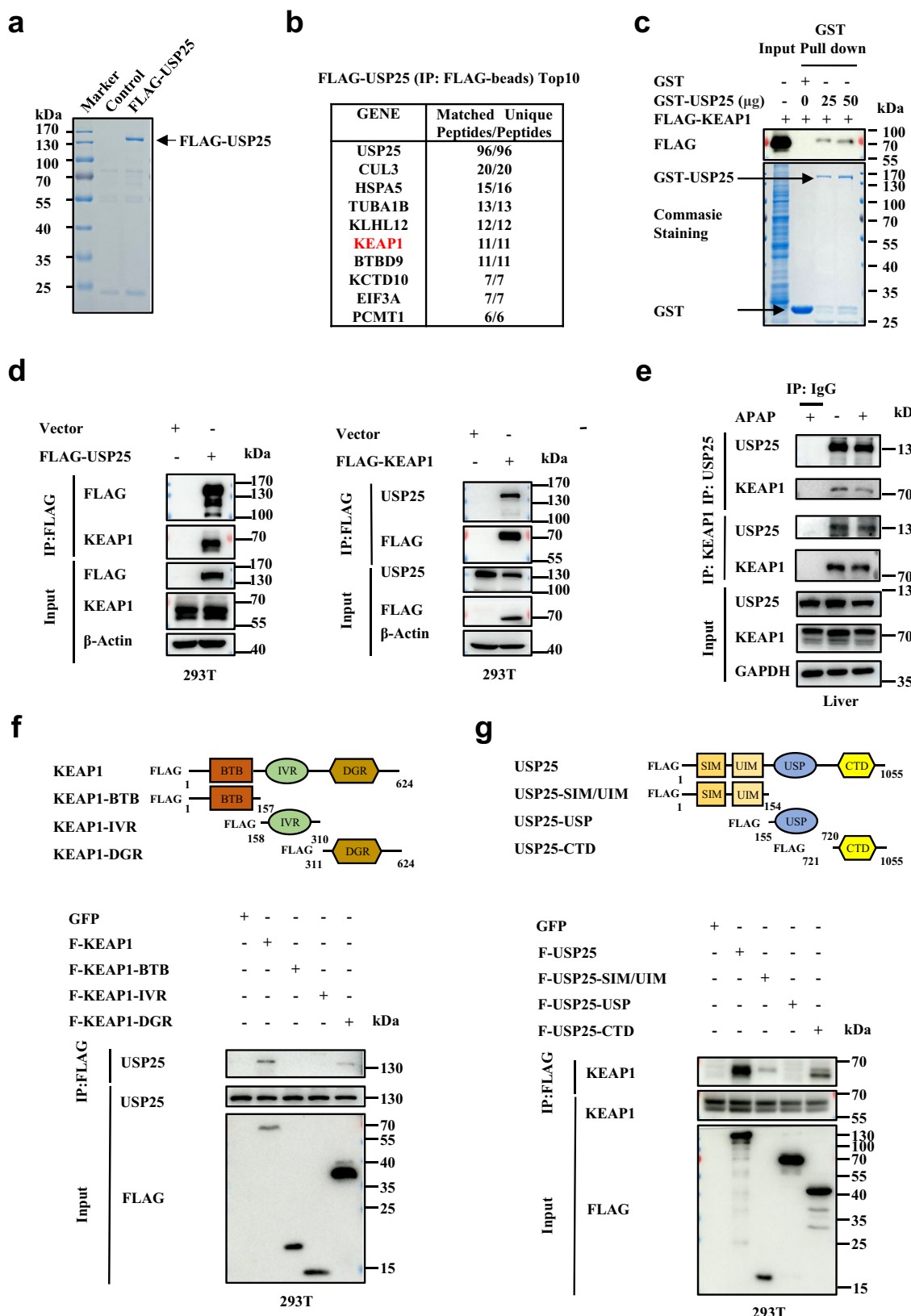

After the planned treatment, the mice were anesthetized, blood samples were collected from the retroorbital venous plexus for testing, and the livers were collected for analysis. The collected whole blood was centrifuged at 3000 g at 4 °C for 10 min to obtain serum. The serum levels of alanine aminotransferase (ALT) and aspartate aminotransferase (AST) were determined with commercial

kits (Sigma-Aldrich, Shanghai, China). The liver tissues were fixed in 4% paraformaldehyde, dehydrated and embedded in paraffin. Sections (5 μm) of the liver tissues were cut and collected. For histological examination, the sections were stained with hematoxylin and eosin (H&E). The necrotic areas in H&E-stained sections were encircled and measured with Image J. 3 view fields under a 10x objective of

**Fig. 3 | USP25 interacts with KEAP1. a** The identification of potential substrates of USP25 by mass spectrometry. FLAG-tagged *USP25* was overexpressed in HEK293T cells, immunoprecipitated, separated with SDS-PAGE. The gel was stained with Commassie blue. **b** Proteins in the gel of (**a**) were identified with mass-spectrometry. Top 10 most abundant proteins identified were listed. **c** In vitro GST pull-down analysis of the interaction between KEAP1 (FLAG- KEAP1) and USP25 (GST-USP25). Arrows indicate the Coomassie blue staining of GST and GST-USP25. **d** Immunoprecipitation analysis of the interaction between FLAG-USP25 and endogenous KEAP1 as well as FLAG- KEAP1 and endogenous USP25 in HEK293T cells (*n* = 3 biologically independent experiments). **e** Co-immunoprecipitation analysis of the interaction between USP25 and KEAP1 in the liver from WT mice treated with or without APAP (300 mg/kg, i.p.) for 6 h (*n* = 3 biologically independent experiments). **f** Mapping the domains in KEAP1 that interact with USP25. FLAG-tagged BTB, IVR, and DGR domains of KEAP1 were expressed in HEK293T cells and immunoprecipitated. The immunoprecipitated were then analyzed for the presence of USP25 (*n* = 2 biologically independent experiments). **g** Mapping the domains in USP25 that interact with KEAP1. FLAG-tagged SIM/UIM, USP, and CTD domains of USP25 were expressed in HEK293T cells and immunoprecipitated. The immunoprecipitated were then analyzed for the presence of KEAP1 (*n* = 2 biologically independent experiments). Source data are provided as a Source Data file.

each section were examined and measured to obtain the percentage of necrotic areas. For ROS measurements, the liver tissue was quick-frozen and sectioned. The sections were then stained with 20 μM dihydroethidium (DHE) (Sigma-Aldrich) and 300 nM 4′,6-diamidine-2′-phenylindole dihydrochloride (DAPI) (Sigma-Aldrich), and micro-imaged under a fluorescence microscope (NIKON, Tokyo, Japan). Glutathione (GSH) were determined with commercial kits (Sigma-Aldrich). For APAP-cysteine (APAP-CYS) concentration measurement, the supernatant of quick-frozen liver tissues after grinding was collected. The concentration was determined using high-performance liquid chromatography (UltiMate 3000 RS, Thermo Fisher Scientific, Waltham, USA) coupled with tandem mass spectrometry (TSQ Quantum, Thermo Fisher Scientific) with the use of standard samples[34].

## Cell culture

HepG2 (HB-8065) and AML12 (CRL-2254) cell lines were obtained from American Type Culture Collection (ATCC, Manassas, VA, USA). HEK293T (GNHu43) cell line were purchased from the Cell Bank of Type Culture Collection of Chinese Academy of Sciences (Beijing, China). HepG2 and HEK293T cells were cultured in Dulbecco's modified Eagle's medium (DMEM), while AML12 cells were cultured in F12/DMEM, supplemented with 10% fetal bovine serum (FBS) and 1% penicillin-streptomycin (Pen-Strep) in a humidified atmosphere at 37 °C and 5% $CO_2$. DMEM, F12/DMEM, FBS, and Pen-Strep were all purchased from Gibco (Grand Island, New York, USA).

For primary hepatocytes, the mouse liver was perfused with 0.05% collagenase type IV (Sigma-Aldrich) to obtain primary hepatocytes, which were plated in six-well plates in DMEM with 10% FBS and 1% Pen-Strep for 4 h for attachment. Subsequently, the cells were maintained in F12/DMEM with 1% Pen-Strep overnight and treated with 10 mM APAP for 0, 3, or 6 h before being harvested for analysis.

To treat the cells with CT1113, the compound was dissolved in dimethyl sulfoxide (DMSO) at a concentration of 10 mM and diluted with cell culture medium to reach final concentrations of use.

## Plasmids, lentiviruses, and adenoviruses

pLKO.1 was used to generate the lentiviruses carrying shRNAs. The shRNA sequences targeting human and mouse *USP25* are listed in Supplementary Table 3. The human USP25 cDNA (WT or C178S mutant) was cloned into the vector pHAGE with a HA tag. The human USP25 and KEAP1, full length or truncated, were subcloned in pCDH with a Flag tag. The sequences of cloning primers are listed in Supplementary Table 4. WT Ub and its mutants (K6, K11, K27, K29, K33, K48, and K63) were subcloned in pCMV with a HA tag.

The lentiviruses for expression of shRNAs or exogenous genes were packaged in 293 T cells with standard packaging plasmids and method. Lentiviral infection of the cells was performed with standard method and the infected cells were selected with 4 μg/ml puromycin (InvivoGen, California, USA) for 2 days to obtain stable gene-knocking down or overexpression cell lines.

For short term gene silencing, we used siRNAs designed and synthesized by GenePharma (Suzhou, China). The siRNAs were transfected into cells with Lipofectamine 2000 (Invitrogen) following the manufacturer's protocol.

The adenoviruses used were provided by GeneChem (Shanghai, China). *Usp25* WT and *Usp25* KO mice were injected with 100 μl of AdshNC ($1 \times 10^9$ pfu) or AdshNRF2 ($1 \times 10^9$ pfu) via the tail vein 5 days before the APAP treatment.

## Western blotting, immunoprecipitation (IP), and GST pull-down

For western blotting analysis of proteins, the cells or tissues were lysed in RIPA buffer (Applygen Technologies Inc., Beijing, China) supplemented with a protease inhibitor cocktail (Roche Diagnostics, Mannheim, Germany), and the lysates were centrifuged at high speed to remove insoluble debris. The protein concentration of the resultant lysates was determined with a bicinchoninic acid (BCA) assay kit (Beyotime, Shanghai, China). Equal amounts of proteins were boiled for 5 min in 5x SDS loading buffer (Biosharp, Hefei, China), separated in an SDS-polyacrylamide gel, and transferred onto nitrocellulose membranes. The membranes were incubated for 1 h in blocking buffer (5% non-fat dry milk in TBST) and then with primary antibodies at 4 °C overnight. After 3 washes with TBST, the membrane was incubated for 1 h at room temperature with horseradish peroxidase (HRP)-conjugated secondary antibodies. The membrane was then washed three times and visualized with SuperSignal™ West Pico Chemiluminescent Substrate (Thermo Fisher Scientific). The expression of GAPDH, β-Actin, or Tubulin was routinely used as a loading control.

For protein half-life analysis, HepG2 cells were treated with 100 μg/mL cycloheximide (Cell Signaling Technology, Boston, USA), collected at different time points, and lysed in 2× SDS buffer. The cell lysates were analyzed with western blotting.

For analysis of proteins in cytoplasm and nuclei, the cytoplasmic and nuclear fractions were prepared with a kit (APPLYGEN, Beijing, China). Briefly, adherent HepG2 cells were washed twice with PBS, and scraped off (100 μL CEA-A buffer per $10^6$ cells) into a plastic tube. The cells were placed on ice and vortexed for 30's every 5 min for 3 cycles, which was followed by centrifuged at 12,000 g for 5 min. The supernatants were the cytoplasmic fraction and the sediments were the nuclear fraction. The nuclear fraction was washed twice with a CEA-A and CEA-B mixed solution, resuspended in NEB buffer and incubated on ice for 30 min with 15's vortex every 10 min, and centrifuged again at 12,000 g for 10 min. The supernatants were the nuclear fraction and proceeded to western blotting analysis together with the cytoplasmic fraction.

For immunoprecipitation, the cells were lysed in NETN buffer and centrifuged to remove debris. The supernatants were cell lysates. The desired protein was precipitated with appropriate antibodies conjugated directly to Sepharose beads (such as Flag M2 beads, Sigma) or via protein A/G-conjugated Sepharose beads. About 2 mg total protein worth of cell lysates were incubated with the antibody-conjugated beads for 1 h at room temperature or overnight at 4 °C. After the

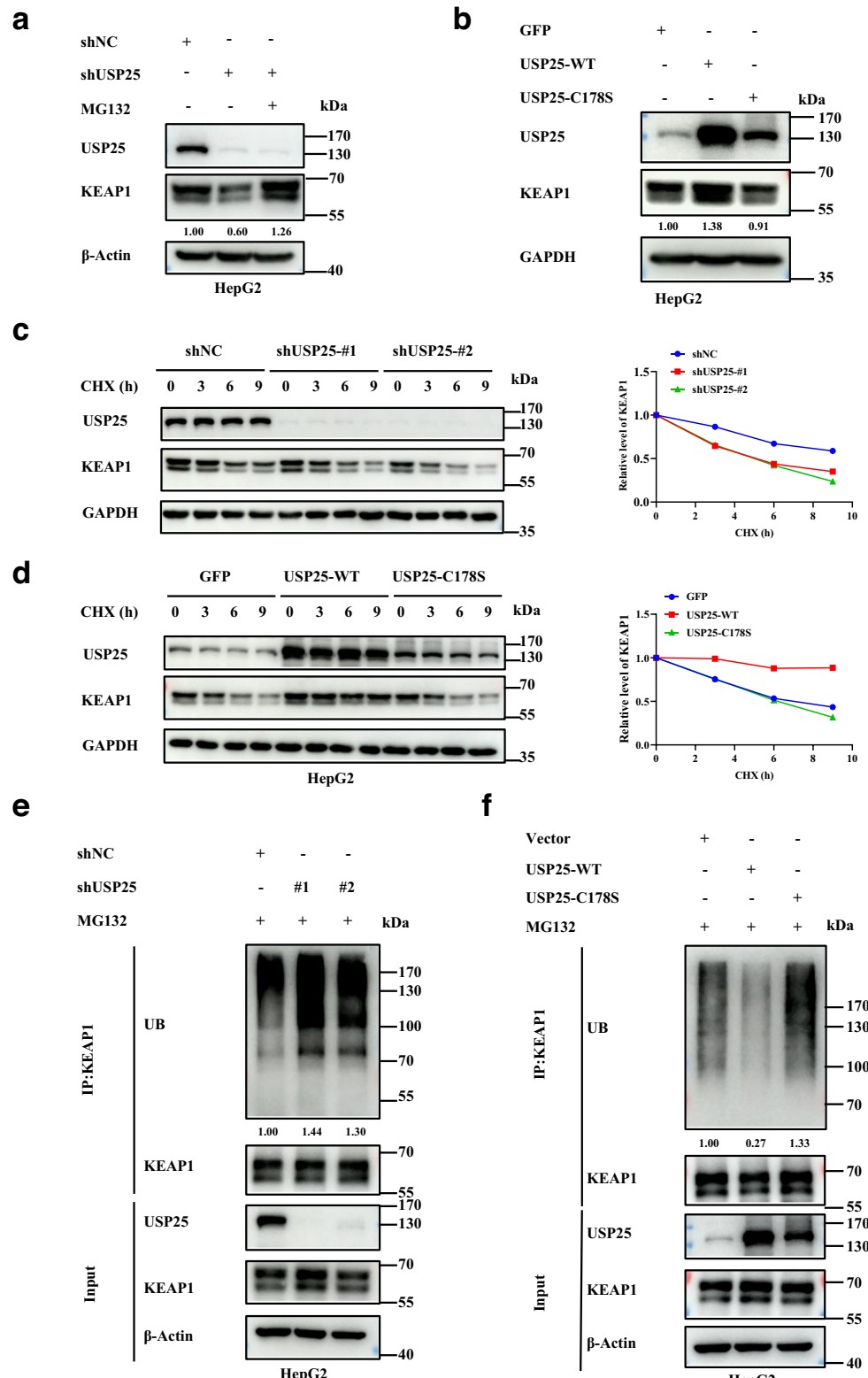

**Fig. 4 | USP25 maintains the stability of KEAP1. a** Western blotting analysis of
KEAP1 in *USP25*-depleted and control HepG2 cells treated with and without the
proteasome inhibitor MG132 (20 µM) for 6 h (*n* = 3 biologically independent
experiments). **b** Western blotting analysis of KEAP1 in HepG2 cells transfected with
GFP-, WT USP25-, or USP25^C178S-expressing plasmids (*n* = 3 biologically independent
experiments). **c** Measurement of the half-lives of KEAP1 in HepG2 cells depleted of
*USP25* expression through cycloheximide chasing (*n* = 2 biologically independent
experiments). **d** Measurement of the half-lives of KEAP1 in HepG2 cells transfected
with GFP-, WT USP25-, or USP25^C178S-expressing plasmids through cycloheximide

chasing (*n* = 2 biologically independent experiments). **e** Western blotting analysis
of the ubiquitinated species of endogenous KEAP1 in HepG2 cells depleted of
*USP25*. The cells were treated with MG132 (20 µM) for 6 h before harvesting for
analysis (*n* = 2 biologically independent experiments). **f** Western blotting analysis
of the ubiquitinated species of endogenous KEAP1 in HepG2 cells transfected with
GFP-, WT USP25-, or USP25^C178S-expressing plasmids. The cells were treated with
MG132 (20 µM) for 6 h before harvesting for analysis (*n* = 2 biologically independent experiments). Source data are provided as a Source Data file.

 9

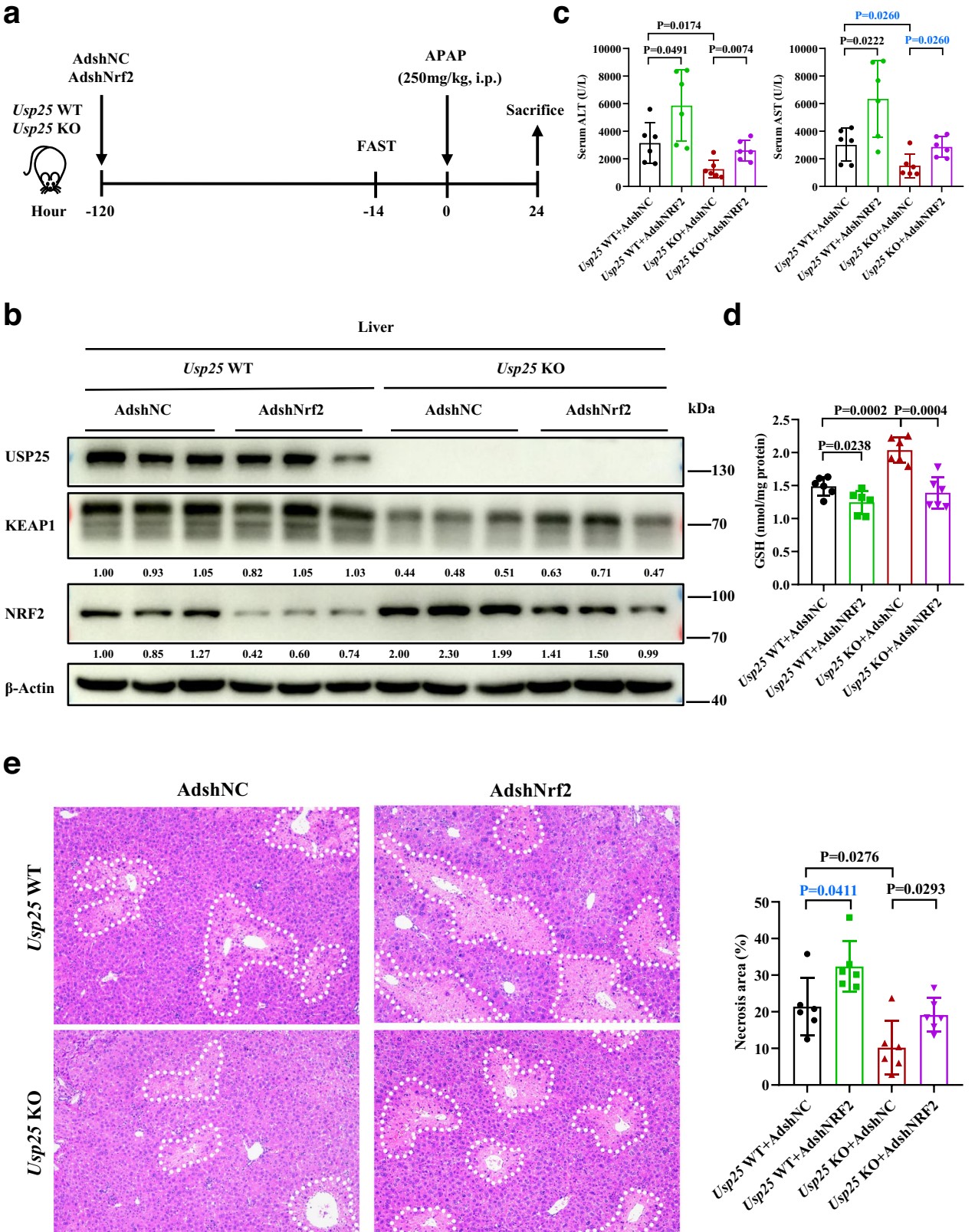

**Fig. 5 | *Nrf2* mediates the protection of APAP-induced liver injury afforded by *Usp25*-deficiency. a** Schematic of the experiment. *Usp25* WT or *Usp25* KO mice were injected with AdshNC or AdshNRF2 via tail vein, followed by administration of 250 mg/kg APAP for 24 h. **b** Western blotting analysis of the indicated proteins in the liver. Samples from 3 animals are shown. The numbers are normalized relative expression of KEAP1 and NRF2. **c, d** Determination of serum ALT, AST (**c**) levels and liver GSH (**d**) levels. *n* = 6 mice per group. **e** Analysis of APAP-induced necrosis in the liver. The liver sections were stained with hematoxylin and eosin and the necrotic areas were encircled and quantified. *n* = 6 mice per group. Scale bar, 200 μm. Error bars denote SEM. Two-tailed student's *t* tests analysis (**c**, **d** and **e**), non-parametric tests with blue colored (**c** and **e**). Source data are provided as a Source Data file.

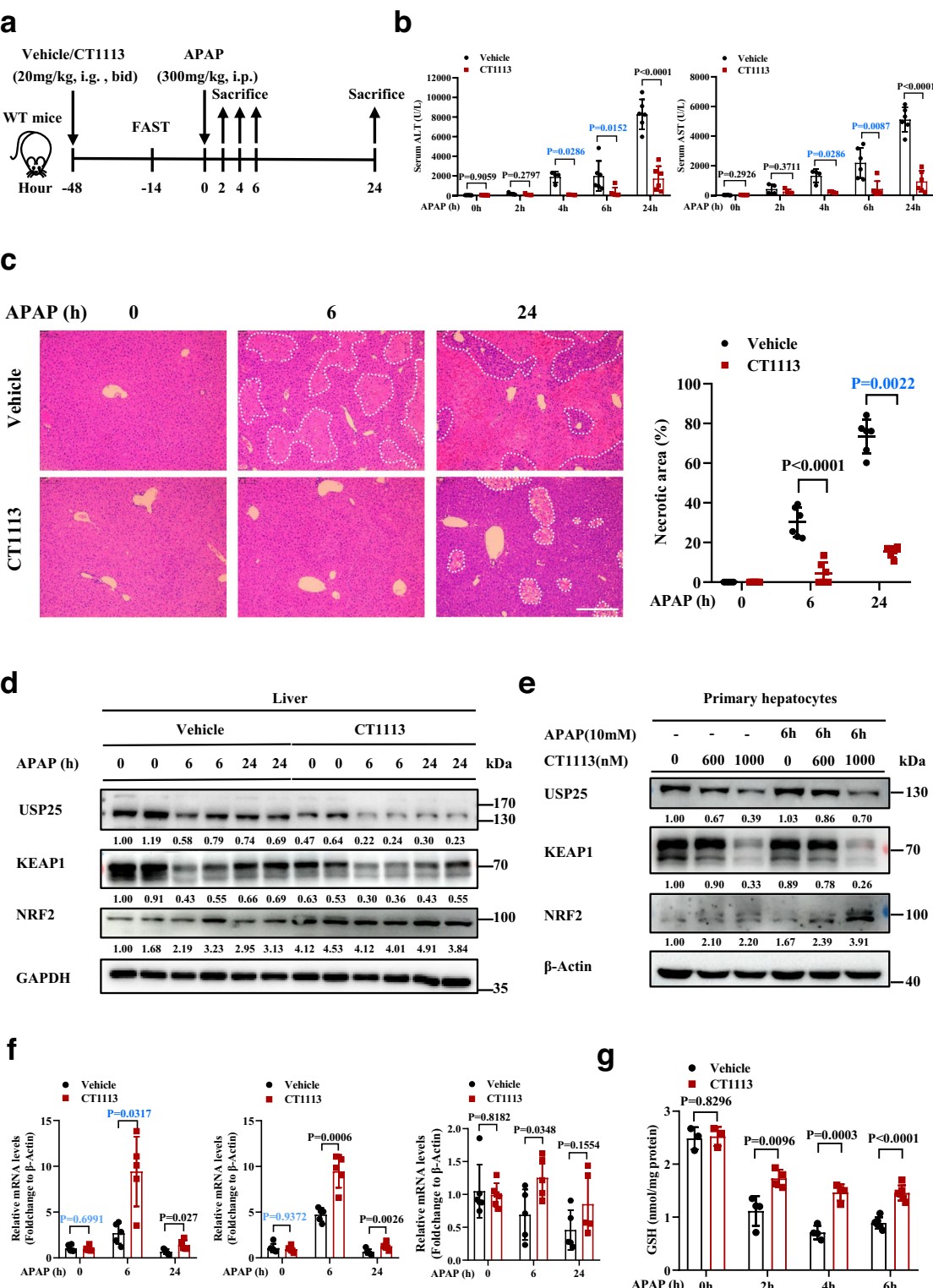

**Fig. 6 | Pharmacologic inhibition of USP25 attenuates APAP-induced liver injury. a** Schematic of the experiment. Male C57BL/6 mice were pre-treated with CT1113 for 2 days (and continued for the rest of time until sacrificing), fasted for 14 h, and then given APAP. The animals were sacrificed at 0, 2, 4, 6 or 24 h after APAP administration. **b** Serum ALT and AST levels. 0 h: $n = 6$ mice, 2 h: $n = 4$ mice, 4 h: $n = 4$ mice, 6 h: $n = 6$ mice, 24 h: $n = 6$ mice. **c** Hematoxylin and eosin staining of the liver sections from mice in (**a**). Necrotic areas were encircled and quantified. $n = 6$ mice per group. Scale bar, 200 μm. **d** Western blotting analysis of the indicated proteins in the liver from the mice in (**a**). **e** Western blotting analysis of the

indicated proteins in the primary hepatocytes isolated from WT mice ($n = 2$ biologically independent experiments). **f** Relative mRNA levels of NRF2 target genes in the liver of the mice in (**a**). 0 h: $n = 6$ mice, 6 h: $n = 5$ mice, 24 h: $n = 5$ mice. **g** The levels of GSH in the liver of the mice in (**a**). 0 h: $n = 3$ mice, 2 h: $n = 4$ mice, 4 h: $n = 4$ mice, 6 h: $n = 6$ mice. The numbers under western blotting bands represent normalized relative expression. Error bars denote SEM. Two-tailed student's $t$ tests analysis (**b**, **c**, **f** and **g**), non-parametric tests with blue colored (**b**, **c** and **f**). Source data are provided as a Source Data file.

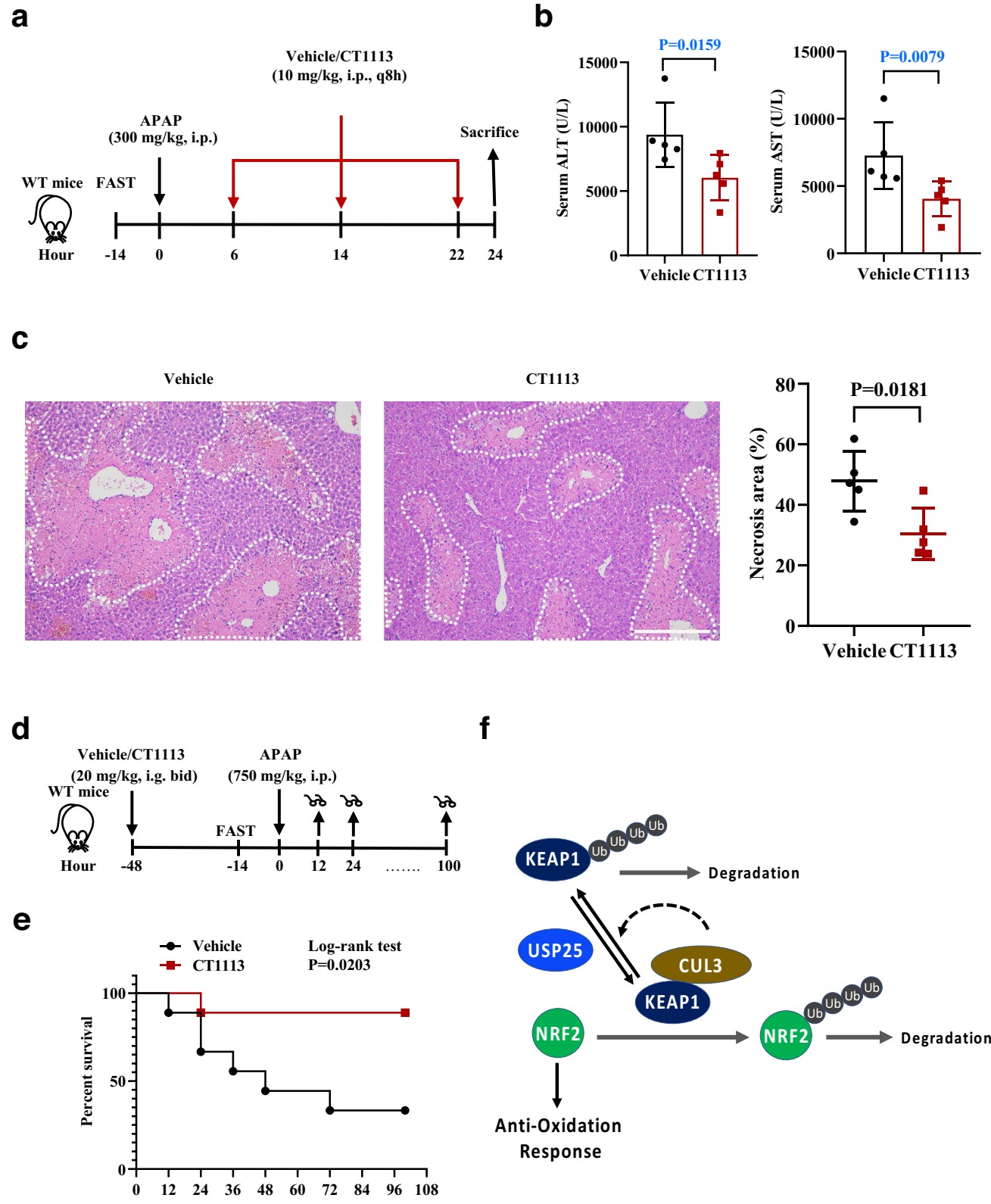

incubation, the beads were washed with NETN buffer at least 3 times. The beads-bound proteins were eluded off through boiling in denaturing SDS-gel loading buffer and analyzed with western blotting.

For GST pull-down experiments, USP25 fused to GST was expressed in *E. coli* and purified with glutathione-agarose beads

(Pharmacia, China). The relevant cell lysates were incubated with approximately 25 μg or 50 μg of purified beads-bound USP25 at 4 °C overnight. The beads were washed five times with GST buffer (1 mM EDTA, pH8.0 tris-HCl, 200 mM NaCl and 1% Triton X-100), and the bound proteins were eluded off the beads with the SDS-gel sample

**Fig. 7 | Alleviating APAP-induced injury and mortality by inhibiting USP25.**
**a** Schematic of the experiment. Male C57BL/6 mice were fasted for 14 h, given APAP (300 mg/kg body weight), and CT1113 6 h later, for a total of 3 times (every 8 h). CT1113 was given through i.p. at half of the oral dose to achieve fast absorption and action. The animals were sacrificed at 24 h after APAP administration. **b** Serum ALT and AST levels. $n = 6$ mice per group. **c** Hematoxylin and eosin staining of the liver sections from mice in (**a**). Necrotic areas were encircled and quantified. $n = 6$ mice per group. Scale bar, 200 μm. **d** Schematic of the experiment. Male C57BL/6 mice were pretreated with CT1113 or vehicle (9 mice per treatment) 48 h before being

fasted for 14 h, given high dose APAP (750 mg/kg body weight). The animals were observed continuously for up to 100 h after APAP administration. CT1113 treatment were continued for the rest of time until death or the end of the experiment. **e** Kaplan-Meier plot of the survival of the mice in (**d**). Error bars denote SEM. **f** Schematic illustration of the role of USP25 and its inhibitor in regulating the KEAP1-NRF2 signaling pathway to oppose APAP-induced liver injury. Error bars denote SEM. Two-tailed student's *t* tests analysis (**c**), non-parametric tests with blue colored (**b**). Source data are provided as a Source Data file.

buffer, separated on an SDS-PAGE gel, and analyzed with western blotting with indicated appropriate antibodies.

The primary and secondary antibodies used in this study are listed in Supplementary Table 1.

## Mass spectrometry analysis
HEK293T cells were transiently transfected with FLAG-USP25 for 48 h, and the cells were harvested and processed for immunoprecipitation with anti-Flag magnetic beads (Bimake), followed by SDS-PAGE. The gel was stained with Coomassie blue and cut into sections to be digested with trypsin. The digested gel fractions were processed for LC-MS/MS analysis on a Thermo Scientific™ Q Exactive Plus mass spectrometer (Thermo Fisher Scientific). Mass spectrometry was performed at Jingjie PTM Biolabs (Hangzhou, China), and the resulting MS/MS data (Supplementary Data 1) were analyzed via Proteome Discoverer v.1.3 (Thermo Fisher Scientific).

## RNA extraction and quantitative real-time PCR (qRT-PCR)
Total RNA was extracted from cells or tissues with TRIzol (Invitrogen, California, USA) according to the manufacturer's protocol. The RNA concentration was determined with Nanodrop 2000 (Thermo Scientific), and 1 μg RNA was reverse transcribed with a kit (Takara, Kyoto, Japan). qRT-PCR was performed in SYBR Green PCR Master Mix (Takara) with QuantStudio 5 (Applied Biosystems, Massachusetts, USA). All mRNA transcript levels were normalized to that of the gene encoding *β*-actin. The sequences of the primers used are listed in Supplementary Table 2.

## Ubiquitination assay
HepG2 or 293 T cells expressing various exogenous proteins of interests or shRNAs were treated with 20 μM MG132 for 6 h. The cells were harvested and lysed in NETN buffer (pH8.0 tris-HCl, 100 mM NaCl, 1 mM EDTA, 0.5% Nonidetp-40) containing 1% SDS and 1% sodium deoxycholate, vortexed vigorously for 15–30 min, boiled for 10 min. Then, 5–9 times of the volume of more NETN buffer were added to reduce SDS content to 0.1% and so produced cell lysates were incubated with appropriate antibody-conjugated beads followed by the rest of immunoprecipitation procedures.

## Quantification and statistical analysis
The gray scale analysis of protein bands, the ROS fluorescence intensity, and the areas of necrosis in liver tissues were calculated with Image J (National Institutes of Health, Maryland, USA). If the data of two groups are normally distributed and have an equal variance, unpaired two-tailed student's *t* tests will be used. Otherwise, non-parametric tests will be used. For survival experiments, the survival curves were generated using the Kaplan-Meier plot and analyzed with log-rank analysis. All statistical analyses were conducted using GraphPad Prism 8 software (GraphPad Software, San Diego, USA), and a *P* value of <0.05 was considered statistically significant.

## Reporting summary
Further information on research design is available in the Nature Portfolio Reporting Summary linked to this article.

## Data availability
The authors declare that all data supporting the findings described in this paper are available in the article and in the Supplementary Information and from the corresponding author upon request. Source data are provided with this paper.

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

## Acknowledgements

We thank Dr. Chen Dong at Tsinghua University and Dr. Bo Zhong at Wuhan University for providing the Usp25 knockout mice. We also thank the core facilities in the Zhejiang Provincial Key Laboratory of Pancreatic Diseases for providing flow cytometry services and experimental support. The authors thank people in Zhang lab for helpful discussions and suggestions throughout the work. This work was supported by grants from the National Key R&D Program of China (grant # 2018YFA0507500) and the National Natural Science Foundation of China (grants # 81600458).

## Author contributions

Conceptualization: C.C., P.Z., F.J. and J.W.; methodology: C.C. (in vitro) and H.M. (in vivo); C.C. (Biochemistry); J.P. and X.Z. (FACS); X.S. (compound synthesis); C.Y. (statistical analysis); writing-original draft: C.C.; writing-review and editing: P.Z. and J.W.; supervision: P.Z., F.J. and J.W.; and funding acquisition: P.Z. and J.W.

## Competing interests

The authors declare no competing interests.
