## [Peer Review File · Nature Communications]

USP25 Regulates KEAP1-NRF2 Anti-Oxidation Axis and Its Inactivation Protects Acetaminophen-Induced Liver Injury in Male MiceReviewers' comments:

Reviewer #1 (Remarks to the Author):

This manuscript by Cai et al. provides a compelling analysis of the heretofore undescribed role of USP25, a de-ubiquitinase, on the turnover of KEAP1, a major repressor of NRF2 signaling. The authors utilize multiple, complementary approaches to define USP25 interactions with KEAP1 and the downstream effects of pathway responses. The inclusion of an APAP challenge model to induce hepatotoxicity known to be modified by KEAP1/NRF2 status exacerbated with NRF2 depletion and protected by amplified KEAP1 expression) provides a wonderful means to evaluate the role of USP25 to modify this pathway in vivo. It is clear that USP25 effects are mediated through KEAP1 and hence NRF2. While the overall experimental approach is quite comprehensive, there is some sloppiness in the presentation/portrayal of the data that needs to be rectified. There are also additional experiments to be considered to buttress some conclusions.

Are there any publications describing the Usp25 and Usp28 knockout mice? How were these genes disrupted and what sort of phenotypes, if any, are manifested?

The use of a pharmacological inhibitor CT1113 of USP25 (and USP28) provides indirect evidence for the primacy of USP25 in protection against APAP hepatotoxicity by virtue of companion studies in which protection is observed in Usp25, but not Usp28 knockout mice. It would be useful to examine this matter more directly by evaluating activation of the NRF2 pathway in the liver and cells of Usp28 disrupted mice and cells, as done in Figure 2. Are levels of KEAP1, NRF2 and downstream target genes equivalent in Usp28 WT and knockout liver, unlike the situation with Usp25 WT and knockout comparisons? Does USP28 lack KEAP1 DUB activity as opposed to the effect of USP25 on this process (Fig S4)? Does CT1113 treatment rescue the increased sensitivity of Usp28 knockout mice to APAP as might be predicted if USP 25 is the key pharmacologic target?

Usp25 genotype does not influence JNK activation, CYP2E1 expression or formation of APAP protein adducts, early events that could affect APAP toxicity. Activation of KEAP1-NRF2 signaling also triggers rapid responses as highlighted in Fig 2 with increased GSH, SOD transcripts at 6 h, but most robust responses (Usp25 null >> Usp WT) not occurring (measured) until 24 h after APAP challenge. Are these changes in gene expression sufficient to account for a protective mechanism or are the other functions (anti-inflammatory?) elicited by NRF2 possibly important? Given the somewhat slowly developing kinetics of gene expression in the NRF2 pathway, is inhibition of DUB with a 1 hr post-APAP challenge any more effective than treatment with a classic electrophilic NRF2 inducer?

The labeling of Fig. 2 is out of sync with the caption and the Results text. There does not appear to be a panel F. Panel H (or I): are these done in HepG2 cells (figure) or 293T cells (legend)?

The results described on p. 10 don't entirely match to the referenced Fig 4A. ... "we depleted USP25 via 2 different shRNAs or overexpressed the DUB in HepG2 cells as well as in AML12 cells. The knockdown of USP25 significantly reduced the protein levels of KEAP1 (Fig 4A, S4A and B)". Only one shRNA is shown; the diminution is more "modest" than "significant" for this N=1 experiment; and no data are shown for AML12 cells.

There are also major discrepancies in the presentation of the data for Fig 7 between the text (p. 13), the figure (panel A) and the figure legend. The text refers to administration of CT1113 one hour after APAP challenge with follow-up doses 10 and 20 hours after that. The figure legend states CT1113 was given "one hr later for the next 2 days". Panel A in the figure indicates that CT1113 was given 48 hours before APAP, and by i.g. administration, not i.p. as indicated in the other forums.

Reviewer #2 (Remarks to the Author):

In this manuscript, authors investigated the role of the deubiquitinase USP25 in regulating KEAP1

ubiquitination and stability on NRF2 activation in acetaminophen (APAP)-induced liver injury in a mouse model. While to investigate USP25 in regulating NRF2 in APAP induced liver injury is novel and may have potential clinical relevance, the study suffered several major weaknesses. The use of these wildtype mice as control for USP25 knockout (KO) mice was questionable. There were concerns on the data rigor. Authors largely ignored the potential effects of USP25 KO or its pharmacological inhibitor on the metabolism of APAP. One of the key notions in this study was that USP25 deficiency may increase NRF2 and thus hepatic GSH synthesis and recovery, but this is only addressed superficially and not in an extensive manner. Overall, the weaknesses seem to overshadow the strengths.

Major concerns:

1. It is well known that the mouse strain is critical to the sensitivity of APAP-induced liver injury. Authors stated that Usp25 KO mice (whole body KO) were obtained from another lab (Dr. Dong Chen) of a different institute. However, wildtype (WT) mice were purchased from a commercial source. This is a major issue as these WT mice may not be comparable to the Usp25 KO mice. The good control mice should be generated from Usp25^{+/-} cross with Usp25^{+/-} mice.
2. APAP metabolism via P450 enzymes such as Cyp2e1 to generate NAPQI and APAP-adducts are critical to APAP-induced mitochondrial damage and necrosis. However, this important issue was not addressed. There were no data to show the levels of Cyp2e1 and APAP-adducts in the WT vs Usp25^{-/-} mice. Similarly, there were no data to show whether pharmacological inhibition of Usp25 by CT1113 would affect hepatic Cyp2e1 and APAP-adducts.
3. Figure 1C and 2A showed decreased USP25 at 6 and 24 hr to similar levels after APAP treatment, why the changes of KEAP1 levels varied and levels of Nrf2 did not increase? Figure 2C, hepatic GSH levels were measured from untreated mice? GSH measurement should be done in a more systemic manner rather than only one time point. Early time point before 6 hrs would be more meaningful to determine the metabolism of APAP.
4. Figure 2E, data from HepG2 cells are irrelevant as HepG2 cells cannot metabolize APAP. Figure 2I, why the pattern of KEAP1 looks more ubiquitinated (smear bands) in control cells than shRNA Usp25 transfected cells?
5. Figure 6, the levels of hepatic GSH, APAP-adducts and Cyp2e1 after CT1113 treatment should be measured. This is very critical to rule out the potential interference of CT1113 on APAP metabolism.
6. Figure 7A the diagram did not match the text in the figure legend. "Next 2 days" treatment would be 48 hours but the data only showed 24 hours. This raised concerns on the data rigor and reproducibility. Moreover, 1 hour is not longer enough. To test the therapeutic effects of CT1113, authors should give CT1113 6 hours post APAP treatment, which would be more relevant to the clinic (as the mice established the injury at 6 hours but not 1 hour).

Reviewer #3 (Remarks to the Author):

Cai et al in this paper describe the role of the deubiquitinating enzyme USP25 in KEAP1 deubiquitination and the prevention of liver injury in mice. The authors use cellular and mouse models in order to identify the regulatory mechanism of KEAP1-NRF2 axis by USP25.

This study is really interesting and well organized, and the authors use a variety of techniques in order to define how USP25 regulates the stability of KEAP1 and how this regulation could be beneficial after APAP-ALI liver injury.

The introduction provides sufficient background and includes all relevant references. The authors knocked down USP25 in cells (using si- and shRNA) and also performed analysis using Usp25 KO mice. They provide all the evidence that KEAP1 is a substrate of the USP25 performing all the necessary experiments (coIPs, ubiquitinating assays, proteomic analysis and biochemical studies in order to study the protein half-life as well as the proteasomal degradation). They also examined the

effect of USP25 inactivation in the prevention and therapeutic approach of the liver injury in mice. Furthermore the conclusions that the authors made are supported by the results.

However, in this present form, this paper has some major concerns that need to be addressed in order to be properly peer-reviewed:

- Figure 7 is missing. Please provide
- Figure S3 is missing. Please provide
- Figure 3 is a bit confusing when you read the results sections. Figure 3D is the actual Figure 3E and Figure 3E is 3D. Please correct it appropriately.
- In Figure 2 there is no panel F. Please correct appropriately.
- The WB need to be quantified in all the cases that the authors say increase or decrease. This is applicable in almost all the Figures. For example: line 106: "protein decreased accordingly" and Figure 1C. The same for the Figures 2, 5 and 6.
- Figure S5A: Please provide quantification of the inhibition and a graph showing the dose response.
- In the quantification and statistical analysis in the method section, the authors state "the gray scale of protein bands" (line 492), but there isn't any graph with the WB quantification. Please provide all the required quantifications.
- Statistical analysis: The authors used the unpaired t-test for their analysis. However, they need to provide all the relevant information that this test is the appropriate: the samples need to follow the normal distribution and have equal variances. Otherwise, an alternative non-parametric test need to be used. The authors need to provide this information regarding the above tests that they used before using the unpaired t-test.

There are some minor comments that the authors could address:

- Please provide all the uncropped WB images.
- There are some minor spelling and grammatical errors. Please revise the whole text for the correct use of English language.
- Please revise all the images to be in the same format. For example in Figure 2G keep all the proteins in capital letters.
- Figure 6E: How many independent experiment were done?

Point-to-point rebuttal:

Reviewer #1:

Are there any publications describing the *Usp25* and *Usp28* knockout mice? How were these genes disrupted and what sort of phenotypes, if any, are manifested?

Thanks for the comment. Yes, both *Usp25* and *Usp28* knockout mice have been reported. They are generally healthy. *Usp25* KO was generated through a gene-trap strategy, which was used in this study as well as in many previous studies which showed that *Usp25* plays a role in inflammation and anti-viral immunity (1-5). Conditional *Usp28* KO (generated through conventional ES cell technology) was first reported in 2014 by Diefenbacher et al (6). The authors found that the loss of *Usp28* was compatible with embryonic development and the KO mice did not display any discernible gross phenotypes, but the KO mice were resistant to carcinogenesis in the intestine induced by genetic inactivation of *Apc*. We also generated a straight knockout allele of *Usp28* through CRISPR gene editing (7). Homozygosity of this allele is compatible with development and postnatal life in mice. The mice were used in this study.

The use of a pharmacological inhibitor CT1113 of USP25 (and USP28) provides indirect evidence for the primacy of USP25 in protection against APAP hepatotoxicity by virtue of companion studies in which protection is observed in *Usp25*, but not *Usp28* knockout mice. It would be useful to examine this matter more directly by evaluating activation of the NRF2 pathway in the liver and cells of *Usp28* disrupted mice and cells, as done in Figure 2. Are levels of KEAP1, NRF2 and downstream target genes equivalent in *Usp28* WT and knockout liver, unlike the situation with *Usp25* WT and knockout comparisons? Does USP28 lack KEAP1 DUB activity as opposed to the effect of USP25 on this process (Fig S4)?

Thanks for the comment. We have now looked at this. We extracted proteins from the liver tissues of *Usp28* WT and *Usp28* KO mice, as well as HepG2 cells with *USP28* depleted. Immunoblotting results showed that *Usp28* deficiency affected the protein levels of neither KEAP1 nor NRF2, both in the liver tissues and the HepG2 cells (Figure S3E and F, new). Further, we now show that *Usp28* deficiency does not alter the expression of downstream target genes of NRF2 (Figure S3D, new) nor the ubiquitination levels of KEAP1 (Figure S3G, new), which is expected and in contrast to *Usp25* deficiency.

Does CT1113 treatment rescue the increased sensitivity of *Usp28* knockout mice to APAP as might be predicted if USP 25 is the key pharmacologic target?

Yes, CT1113 does provide a rescue (Figure S5D, new).

Usp25 genotype does not influence JNK activation, CYP2E1 expression or formation of APAP protein adducts, early events that could affect APAP toxicity. Activation of KEAP1-NRF2 signaling also triggers rapid responses as highlighted in Fig 2 with increased GSH, SOD transcripts at 6 h, but most robust responses (*Usp25* null >> *Usp*

WT) not occurring (measured) until 24 h after APAP challenge. Are these changes in gene expression sufficient to account for a protective mechanism or are the other functions (anti-inflammatory?) elicited by NRF2 possibly important?

Thanks for the comment. We could not rule out the possibility that the other functions elicited by NRF2 may also contribute to the protection in *Usp25* KO mice. These changes of gene expression we revealed only serve as an indication that NRF2 is activated.

Comment 4: Given the somewhat slowly developing kinetics of gene expression in the NRF2 pathway, is inhibition of DUB with a 1 hr post-APAP challenge any more effective than treatment with a classic electrophilic NRF2 inducer?

Thanks for the comment. We have compared side-by-side, CT1113 and bardoxolone methyl (RTA402), an NRF2 inducer. Apparently, the two agents have similar efficacies against APAP-induced liver damage (Figure S7 A-C, new).

The labeling of Fig. 2 is out of sync with the caption and the Results text. There does not appear to be a panel F. Panel H (or I): are these done in HepG2 cells (figure) or 293T cells (legend)?

We apologize for the mistake. The experiment was done on HepG2 cell. It is corrected now.

The results described on p. 10 don't entirely match to the referenced Fig 4A. ..."we depleted USP25 via 2 different shRNAs or overexpressed the DUB in HepG2 cells as well as in AML12 cells. The knockdown of USP25 significantly reduced the protein levels of KEAP1 (Fig 4A, S4A and B)". Only one shRNA is shown; the diminution is more "modest" than "significant" for this N=1 experiment; and no data are shown for AML12 cells.

Thanks for the comment and we are sorry for the confusion. Although Fig. 4A only showed one shRNA, two were shown for Fig. S4A and B. We have made that clear in the revised manuscript. The data from AML12 cells were shown in Figure S4B.

We also tuned down the wording, by removing "significantly". At the same time, we quantitated the band intensities to help comprehend the data.

There are also major discrepancies in the presentation of the data for Fig 7 between the text (p. 13), the figure (panel A) and the figure legend. The text refers to administration of CT1113 one hour after APAP challenge with follow-up doses 10 and 20 hours after that. The figure legend states CT1113 was given "one hr later for the next 2 days". Panel A in the figure indicates that CT1113 was given 48 hours before APAP, and by i.g. administration, not i.p as indicated in the other forums.

Thanks for the comment and we apologize for the discrepancies in presenting the data. We have now corrected that. Apparently, the part of the Fig.6 legend was copied for Fig. 7.

In addition, as suggested by another reviewer, we redid the experiment with CT1113

given at 6 hr (instead of originally 1 hr) after APAP treatment to really show if CT1113 could be therapeutically effective. The results (Figure 7A-C, new) showed that it could.

Reviewer #2

It is well known that the mouse strain is critical to the sensitivity of APAP-induced liver injury. Authors stated that *Usp25* KO mice (whole body KO) were obtained from another lab (Dr. Dong Chen) of different institute. However, wildtype (WT) mice were purchase from a commercial source. This is a major issue as these WT mice may not be comparable to the *Usp25* KO mice. The good control mice should be generated from *Usp25*^{+/-} cross with *Usp25*^{+/-} mice.

Thanks for the comment and sorry for the confusion. We certainly did not purchase wildtype mice to pair with *Usp25* (or *Usp28*) knockout mice. Whenever knockout mice were used, the control were wildtype littermates! This is a common practice in the lab. Commercial wildtype mice (C57BL/6) were only used in pharmacological treatment experiments. We have made it clear in the revised manuscript.

APAP metabolism via P450 enzymes such as Cyp2e1 to generate NAPQI and APAP-adducts are critical to APAP-induced mitochondrial damage and necrosis. However, this important issue was not addressed. There were no data to show the levels of Cyp2e1 and APAP-adducts in the WT vs *Usp25*^{-/-} mice. Similarly, there were no data to show whether pharmacological inhibition *Usp25* by CT1113 would affect hepatic Cyp2e1 and APAP-adducts.

Thanks for the comments. However, we did perform afore mentioned analyses. The data were included in Figure S3B (on difference in CYP2E1 expression levels between *Usp25* WT and *Usp25* KO) and C (APAP-adducts). As for CT1113 treatment, we examined the expression of CYP2E1 and did not find any effects (Figure S5E, new). Unfortunately, we run out of APAP-adduct standard and could not obtain it from commercial sources anymore. We therefore were unable to analyze the effect of CT1113 on the formation of APAP-adducts.

Figure 1C and 2A showed decreased USP25 at 6 and 24 hr to similar levels after APAP treatment, why the changes of KEAP1 levels varied and levels of Nrf2 did not increase?

Thanks for the comments. The coming back of KEAP1 levels at 24 hr after APAP treatment is likely due to a different and unknown mechanism(s) at play. As for NRF2, our results are consistent with others (8,9), but NRF2 levels did increase in primary mouse hepatocytes in response to APAP (Fig. 2D). It is likely that APAP treatment in mice did not reach a uniform level across all hepatocytes, and only those hepatocytes that have higher levels of APAP would mount responses, which provides a plausible explanation for the discrepancy between the in vivo and in vitro observations.

Figure 2C, hepatic GSH levels were measurement from untreated mice? GSH measurement should be done in a more systemic manner rather than only one time points. Early time point before 6 hrs would be more meaningful to determine the metabolism of APAP.

Thanks for the comments. Figure 2C were GSH measurements at 6 hr after APAP treatment as indicated in the text (but did not specify in the legend, for which we apologize). Per your suggestion, we have now measured GSH levels at 0, 2, 4, and 6 h (new Figure 2C). There were no differences between *Usp25* WT and *Usp25* KO mice. The levels decreased after APAP treatment, but the decreases were much larger in the ko mice than in the control WT mice, a likely result of earlier and stronger NRF2 activation in the ko mice.

Figure 2E, data from HepG2 cells are irrelevant as HepG2 cells cannot metabolize APAP.

Thanks for the comment. We have now removed those data, but kept the part without APAP treatment to show that USP25 can regulate KEAP1 and NRF2 in human cells (new Figure 2E).

Figure 2I, why the pattern of KEAP1 look more ubiquitinated (smear bands) in control cells than shRNA *Usp25* transfected cells?

Thanks for the comment. Here we were not looking at the ubiquitination of KEAP1. The smearing of KEAP1 bands came from overexpression of exogenous KEAP1 (and also overexposure of the blot). We have now redone the experiment and obtained much cleaner results (Figure 2H).

Figure 6, the levels of hepatic GSH, APAP-adducts and Cyp2e1 after CT1113 Treatment should be measured. This is very critical to rule out the potential interference of CT1113 on APAP metabolism.

Thanks for the comments. We have now examined the expression of CYP2E1 after CT1113 treatment and did not find any effects on the expression (Figure S5F, new). Unfortunately, we run out of APAP-adduct standard and could not obtain it from commercial sources anymore. We therefore were unable to analyze the effect of CT1113 on the formation of APAP-adducts.

Comment 6: Figure 7A the diagram did not match the text in the figure legend. “Next 2 days” treatment would be 48 hours but the data only showed 24 hours. This raised concerns on the data rigor and reproducibility. Moreover, 1 hour is not long enough. To test the therapeutic effects of CT1113, authors should give CT1113 6 hours post APAP treatment, which would be more relevant to the clinic (as the mice established the injury at 6 hours but not 1 hour).

Thanks for the comments. We made a mistake in preparing the legend. A part of the Fig.6 legend was copied for Fig. 7. We are terribly sorry for the mistake.

We agree that 1 hour is not long enough for testing the therapeutic effects of CT1113,

and per your suggestion, we redid the experiment with CT1113 given at 6 hr after APAP treatment (Figure 7A-C, new). CT1113 could still show therapeutic efficacies!

Reviewer #3

Figure 7 is missing. Please provide.

We are terribly sorry, but something must have happened over the internet, as two other reviewers did see Fig. 7, and per their suggestion, we have revised Figure 7.

Figure S3 is missing. Please provide.

We are terribly sorry, but again something must have happened over the internet.

Figure 3 is a bit confusing when you read the results sections. Figure 3D is the actual Figure 3E and Figure 3E is 3D. Please correct it appropriately.

We are terribly sorry for this confusion. We have corrected the labelling.

In Figure 2 there is no panel F. Please correct appropriately.

We are terribly sorry for this. We have corrected the labelling.

The WB need to be quantified in all the cases that the authors say increase or decrease. This is applicable in almost all the Figures. For example: line 106: "protein decreased accordingly" and Figure 1C. The same for the Figures 2, 5 and 6.

Thanks for the comment. We have quantitated the band intensities with Image J presented the numbers across the Figures.

Figure S5A: Please provide quantification of the inhibition and a graph showing the dose response.

Thanks for the comment. It is done now.

In the quantification and statistical analysis in the method section, the authors state "the gray scale of protein bands" (line 492), but there isn't any graph with the WB quantification. Please provide all the required quantifications.

Thanks for the comment. It is done now.

Statistical analysis: The authors used the unpaired t-test for their analysis. However, they need to provide all the relevant information that this test is the appropriate: the samples need to follow the normal distribution and have equal variances. Otherwise, an alternative non-parametric test needs to be used. The authors need to provide this information regarding the above tests that they used before using the unpaired t-test.

Thanks for the comments. We redid all statistical analyses. We first conducted tests for data normality. If the data from the analyzed group were normally distributed, we would use unpaired t-test; if not, we would use non-parametric test. If the t-test indicated unequal variances between the two groups of normally distributed data, we

would switch to non-parametric test. These are now included in the method section.

Please provide all the uncropped WB images.

Done.

There are some minor spelling and grammatical errors. Please revise the whole text for the correct use of English language.

Thanks for the comment. It is done now.

Please revise all the images to be in the same format. For example, in Figure 2G keep all the proteins in capital letters.

Thanks for the comment. It is done now.

Figure 6E: How many independent experiments were done?

Thanks for the comment. At each time point, there were more than 3 animals. Each animal accounts for one independent experiment, thus more than 3 independent experiments for each time point.

References:

1. Zhong, B., Liu, X., Wang, X., Chang, S. H., Liu, X., Wang, A., Reynolds, J. M., and Dong, C. (2012) Negative regulation of IL-17-mediated signaling and inflammation by the ubiquitin-specific protease USP25. *Nat Immunol* **13**, 1110-1117
2. Wang, X.-M., Yang, C., Zhao, Y., Xu, Z.-G., Yang, W., Wang, P., Lin, D., Xiong, B., Fang, J.-Y., Dong, C., and Zhong, B. (2020) The deubiquitinase USP25 supports colonic inflammation and bacterial infection and promotes colorectal cancer. *Nature Cancer* **1**, 811-825
3. Liu, Z., Qi, M., Tian, S., Yang, Q., Liu, J., Wang, S., Ji, M., Yu, R., Zeng, S., Li, J., Wei, Y., and Dong, W. (2022) Ubiquitin-Specific Protease 25 Aggravates Acute Pancreatitis and Acute Pancreatitis-Related Multiple Organ Injury by Destroying Tight Junctions Through Activation of The STAT3 Pathway. *Frontiers in Cell and Developmental Biology* **9**
4. Ye, B., Zhou, H., Chen, Y., Luo, W., Lin, W., Zhao, Y., Han, J., Han, X., Huang, W., Wu, G., Wang, X., and Liang, G. (2023) USP25 Ameliorates Pathological Cardiac Hypertrophy by Stabilizing SERCA2a in Cardiomyocytes. *Circulation research* **132**, 465-480
5. Yang, Y., Zhan, X., Zhang, C., Shi, J., Wu, J., Deng, X., Hong, Y., Li, Q., Ge, S., Xu, G., and He, F. (2023) USP25-PKM2-glycolysis axis contributes to ischemia reperfusion-induced acute kidney injury by promoting M1-like macrophage polarization and proinflammatory response. *Clinical immunology (Orlando, Fla.)*, 109279
6. Diefenbacher, M. E., Popov, N., Blake, S. M., Schulein-Volk, C., Nye, E., Spencer-Dene, B., Jaenicke, L. A., Eilers, M., and Behrens, A. (2014) The deubiquitinase USP28 controls intestinal homeostasis and promotes colorectal cancer. *J Clin Invest* **124**, 3407-3418
7. Sun, X., Cai, M., Wu, L., Zhen, X., Chen, Y., Peng, J., Han, S., and Zhang, P. (2022) Ubiquitin-specific protease 28 deubiquitinates TCF7L2 to govern the action of the Wnt

signaling pathway in hepatic carcinoma. *Cancer Sci* **113**, 3463-3475

8. Li, Y., Xu, J., Li, D., Ma, H., Mu, Y., Huang, X., and Li, L. (2020) Guavinoside B from *Psidium guajava* alleviates acetaminophen-induced liver injury via regulating the Nrf2 and JNK signaling pathways. *Food Funct* **11**, 8297-8308
9. Lv, H., Hong, L., Tian, Y., Yin, C., Zhu, C., and Feng, H. (2019) Corilagin alleviates acetaminophen-induced hepatotoxicity via enhancing the AMPK/GSK3beta-Nrf2 signaling pathway. *Cell Commun Signal* **17**, 2

REVIEWERS' COMMENTS

Reviewer #1 (Remarks to the Author):

The authors have done an excellent job of responding the comments and concerns of the reviewers. The additional experiments and text changes provide clarity and enhanced rigor to the study.

Reviewer #2 (Remarks to the Author):

Authors have addressed my concerns.

Reviewer #3 (Remarks to the Author):

The authors have addressed all my comments. After including the comments from the other reviewers as well, now the manuscript has improved a lot, and I think that could be accepted for publication in Nature Communications.

Point-to-point rebuttal:

Reviewer #1:

The authors have done an excellent job of responding the comments and concerns of the reviewers. The additional experiments and text changes provide clarity and enhanced rigor to the study.

Thanks for your comments.

Reviewer #2

Authors have addressed my concerns.

Thanks for your comments.

Reviewer #3

The authors have addressed all my comments. After including the comments from the other reviewers as well, now the manuscript has improved a lot, and I think that could be accepted for publication in Nature Communications.

Thanks for your comments.